# Neural Networks Modeling for Prediction of Required Resources for Personalized Endourologic Treatment of Urolithiasis

**DOI:** 10.3390/jpm12050784

**Published:** 2022-05-12

**Authors:** Clemens Huettenbrink, Wolfgang Hitzl, Sascha Pahernik, Jens Kubitz, Valentin Popeneciu, Jascha Ell

**Affiliations:** 1Department of Urology, Nuremberg General Hospital, Paracelsus Medical University, 90419 Nuremberg, Germany; sascha.pahernik@klinikum-nuernberg.de (S.P.); valentin.popeneciu@klinikum-nuernberg.de (V.P.); jascha.ell@klinikum-nuernberg.de (J.E.); 2Research and Innovation Management (RIM), Team Biostatistics and Publication of Clinical Trial Studies, Paracelsus Medical University, 5020 Salzburg, Austria; wolfgang.hitzl@pmu.ac.at; 3Department of Ophthalmology and Optometry, Paracelsus Medical University Salzburg, 5020 Salzburg, Austria; 4Research Program Experimental Ophthalmology and Glaucoma Research, Paracelsus Medical University, 5020 Salzburg, Austria; 5Department of Anaesthesiology, Nuremberg General Hospital, Paracelsus Medical University, 90419 Nuremberg, Germany; jens.kubitz@klinikum-nuernberg.de

**Keywords:** neural network, personalized, urolithiasis, laser lithotripsy, ureterorenoscopy, planning

## Abstract

When scheduling surgeries for urolithiasis, the lack of information about the complexity of procedures and required instruments can lead to mismanagement, cancellations of elective surgeries and financial risk for the hospital. The aim of this study was to develop, train, and test prediction models for ureterorenoscopy. Routinely acquired Computer Tomography (CT) imaging data and patient data were used as data sources. Machine learning models were trained and tested to predict the need for laser lithotripsy and to forecast the expected duration of ureterorenoscopy on the bases of 474 patients over a period from May 2016 to December 2019. Negative predictive value for use of laser lithotripsy was 92%, and positive predictive value 91% before application of the reject option, increasing to 97% and 94% after application of the reject option. Similar results were found for duration of surgery at ≤30 min. This combined prediction is possible for 54% of patients. Factors influencing prediction of laser application and duration ≤30 min are age, sex, height, weight, Body Mass Index (BMI), stone size, stone volume, stone density, and presence of a ureteral stent. Neuronal networks for prediction help to identify patients with an operative time ≤30 min who did not require laser lithotripsy. Thus, surgical planning and resource allocation can be optimised to increase efficiency in the Operating Room (OR).

## 1. Introduction

Preoperative planning of procedures in surgery requires a lot of prior knowledge and depends on different decisions and influencing factors. In addition to the type of therapy, existing imaging and patient characteristics are important aspects that influence the planning of a procedure. Considering that the main source of income for hospitals is the operating theatre, inefficient planning of the surgical program results in higher costs. The goal of effective surgical planning is to reduce surgical duration, reduce sterilization costs, optimize the use of technology needed for surgery, and achieve therapeutic success to avoid subsequent interventions. Another major problem is the incidence of emergency-related changes in the ongoing surgical program, which makes it necessary to postpone elective surgeries and ultimately results in the cancellation of scheduled surgeries. This leads to patient dissatisfaction. With effective planning of the elective program, involvement of emergency interventions can be more responsive. Furthermore, a more efficient utilization of anaesthesia capacity can be achieved through appropriate planning. This results in better utilization of the available surgical capacity [1]. However, managing the operating room (OR) is a complex task owing to the conflicting priorities and preferences of the stakeholders and scarcity of costly resources [2]. There are simulations and modeling techniques that should result in support for the OR manager. Through decision support systems and computer models, process improvements can be achieved to make the planning processes more effective [3]. Processes such as sequencing operations in order of increasing length, along with factors such as sufficient backup capacity, can counteract the elimination of elective operations and ultimately result in better cost efficiency [4]. For the treatment of urolithiasis, these findings have substantial significance. Since urinary stone disease has the status of a widespread disease with high incidence and rising prevalence, the treatment of urolithiasis accounts for a large proportion of urologic surgeries. At the same time, the majority of surgeries are elective procedures [5]. For example, endourological stone repair using ureterorenoscopy has made much progress in recent years, partly due to miniaturization of the instruments, and the number of operations continues to rise. As an example, the numbers for ureterorenoscopies in Germany increased from 32,203 cases in 2006 to 78,125 cases in 2019 [6]. There are already isolated attempts to simulate the interventions and their duration for better planning. For the treatment of kidney stones using flexible ureterorenoscopy with laser lithotripsy, prediction models have been developed to predict the operation time. It was shown that different parameters, such as the presence of a ureteral stent before surgery, or gender, influence the operation time for flexible ureterorenoscopy endourological [7]. Using different preoperative data, models for calculating the operation time can, thus, already be created for flexible ureterorenoscopy. The more complex the procedure, the more factors naturally play a role in accurately predicting the operating time. It is known that preoperative parameters such as stone density and stone size have an influence on the laser energy used in the case of laser application [8]. Thus, there is also an effect on the operation time due to the duration of the energy applied. However, prediction models regarding the operation time should at best determine the requirement for laser application in advance in order to allow for better planning of the operation. Our goal was to develop a method for assessing semirigid ureterorenoscopic procedures to estimate operative time and predict the use of laser lithotripsy.

## 2. Materials and Methods

From May 2016 to December 2019, 474 patients who underwent semirigid ureterorenoscopy for ureterolithiasis were analysed for the development of a model using clinical imaging data to predict the need for laser lithotripsy and to make a forecast of the expected duration of surgery. The conduct of the study was reviewed and approved by the institutional review board (IRB-2021-029). The inclusion criterion was the presence of Computer Tomography (CT) imaging, either as part of the emergency initial presentation or at time of preoperative presentation. All patients who underwent semirigid ureterorenoscopy with a single stone in the ureter or through semirigid accessible portion of the renal pelvis (pyeloureteral junction) were included. The patients had no previous therapies, such as extracorporeal shock wave therapy, which could have an influence on the stone treatment. Anomalies such as horseshoe kidney or ureter duplex were excluded. Regarding the stone size, there was no preselection. All stones that could be treated by semirigid ureterorenoscopy were included which reflects the clinical reality most appropriately. Patients with nephrolithiasis of the renal pelvis or calyces that could only be treated by flexible ureterorenoscopy were not included due to the different possible stone positions in the renal calyces and the resulting special anatomical influence on the procedure compared to the otherwise standardized semirigid ureterorenoscopy up to the renal pelvis. The aim was to give a preoperative estimate of the duration of surgery in addition to a statement about the necessity of laser use. For this purpose, the question was whether an operation would take more than 30 min or would be completed within that time. Considering the additional time required for anaesthesia, surgical slots of more than one hour are thus unlikely for ureterorenoscopy. This represents a manageable planning unit and provides for the most effective utilization of available operating times. Studies have also shown that for patients with ureterorenoscopy that lasted less than 30 min, postoperative short-term Mono-J insertion is a possible option without the risk of additional complications [9]. All operations were performed by a total of 17 urologists. For reasons of quality assurance, only interventions by specialized urologists were included in the creation of the prediction model. Procedures performed by urologists who had less expertise (<50 cases of semirigid ureterorenoscopy) were excluded from the analysis. Therefore, procedures from 17 surgeons were included in the study allowing a broad representation of clinical urological expertise when creating the prediction model and ensuring the stability of the model. In order to ensure good practicability in daily practice, only variables that can be easily collected and measured in a standardized way were included in the prediction models. Data from available CT imaging concerning stone size, volume, density, and location as well as general patient data for age, size, weight, and medical history regarding ureteral stenting were extensively analysed for their predictive power. In case of preoperative ureteral stenting, this was performed during the emergency first visit due to therapy-refractory colic or septic constellation. All ureterorenoscopies were performed using Wolf^®^ semirigid ureterorenoscopes ranging in size from 4.8 to 8.4 Charrière (Knittlingen, Germany). The laser lithotripsy was performed by using Lumenis^®^ Pulse 120H holmium laser with 532 nm (Yokneam, Israel). Lithotripsy was carried out with standardized fragmentation program at 20 W. If laser lithotripsy was not necessary, a nitinol basket was used for stone removal. To improve the performances of the prediction models, an important and common method in machine learning—the ‘reject option’—was applied, i.e., the predictive models were allowed to refuse to make a prediction (i.e., the model was ‘in doubt’). This implies that if no prediction is made for an individual, the clinical expert must decide whether laser lithotripsy is needed and whether the duration of surgery will be less than 30 min. The advantage of this option is that it increases the accuracy of prediction such that the medical expert can safely rely on the prediction. The disadvantage of this method is that a large number of individuals do not receive a prediction. If no prediction is made by the models, the decisions have to be made by the clinical expert as usual in daily practice.

### 2.1. Statistical Methods

Data were screened and cleaned for incomplete, incorrect, and missing data. Continuously distributed data were analyzed for various distributions including normal, log-normal, and gamma distributions. Fisher’s exact test and Pearson’s chi-square tests were used to analyze cross-tabulation tables. Generalized linear models (GLE) with log-normal and gamma distributions were used for continuous variables. Mann-Whitney U tests and Spearman correlations were used as non-parametric methods.

### 2.2. Machine Learning Algorithms

Multilayer perceptron neural networks, support vector machines, nearest neighbors classifiers, random forest models, and Bayes classifiers were analyzed and their model performances were compared against each other. An overview of pre-processing, model training, split of data into training, validation and test sample, training stop criteria, selection of thresholds for the reject option and final test of the model is given in Appendix A. A description of the network architecture of the neural network including all layers and activation functions is given in Appendix A.

### 2.3. Feature Selection

Feature selection was performed in two steps: in a first step, prior medical expert knowledge together with results of published studies were used to select a large set of possible candidate predictors which are known to have high or moderate prediction power. Intensive discussion between the clinicians and statistician of possible predictor variables ended up with the decision to select 14 candidate predictor variables, which in the second step were then offered to a genetic variable selection algorithm to further reduce the number of predictor variables. The genetic algorithms for feature selection algorithm were directly integrated as part of the learning algorithm (‘embedded method’) which means that the genetic algorithm was applied during training.

### 2.4. Reject Option

To allow the algorithm to refuse a prediction (‘if the prediction model is in doubt’), the ‘reject option’ was applied [10]. This means that, in the finally trained model, two cut-offs instead of one for the posterior probabilities were chosen in the training sample and then tested in the test samples. We aimed to find prediction models with high negative and positive predictive values for two reasons: a false negative prediction will extend time of surgery, or stone lithotripsy has to be cancelled due to the absence of a laser which might be used by other surgery teams. A false positive prediction may lead to delay of preoperative preparation of the patient. In the worst case, this increases cost of laser equipment (e.g., laser probe). In order to achieve high negative and positive predictive values, we checked whether the reject option increased predictive values as compared with models without using a reject option. To explain why a certain number of patients did not receive a prediction, how closely both data clouds (endpoint laser lithotripsy) are stuck together, and thus the need for a reject option, we illustrate the overlap between various data distributions using matrix plots of univariate predictors and PCA (Figure 1a,b).

### 2.5. Model Performance

Finally, to assess model performance after applying the reject option, negative and positive predictive values and the percentage of subjects without a prediction were computed in the training and test samples. Results from the training and test samples were compared and reported to demonstrate how the algorithms generalized to new previously unseen data [10]. All reported tests were two-sided, and *p*-values < 0.05 were considered statistically significant. All statistical analyses in this report were performed using STATISTICA 13 (Hill, T. & Lewicki, P. Statistics: Methods and Applications. StatSoft, Tulsa, OK, USA) and MATHEMATICA 12.1 (Wolfram Research, Inc., Mathematica, Version 12.0, Champaign, IL, USA, 2019) [11,12].

## 3. Results

A total of 474 patients undergoing ureterorenoscopy were included in the study between May 2016 and December 2019, of which data of 73 patients had to be excluded due to incomplete data. In total, data from 401 patients were included for evaluation (Table 1).

Almost half of the ureteral stones were located in the distal ureter. Of the patients treated, the majority had received a ureteral stent prior to ureterorenoscopy, which had been inserted as part of an emergency treatment. Approximately one-fourth of the patients underwent laser lithotripsy with a mean operative time of less than half an hour. Stone size ranged from 1–27 mm with a mean size of 6.58 mm.

### 3.1. Uni- and Multivariate Significant Predictors

The following parameters were identified as univariate predictive factors for the prediction of laser application and an operation duration of ≤30 min: patient age, gender, height, weight, resulting BMI, ureteral stone size (height, width, depth) and resulting spherical volume, stone density in Hounsfield units, and whether a preoperative ureteral stent was present. In multivariable analysis to assess predictors for laser use, patient age (*p* < 0.01, OR 1.02, 95% CI: 1.01–1.04), weight (*p* = 0.04, OR 1.02, 1.004–1.4), height (*p* < 0.01, OR 1.67, 1.49–1.87), width (*p* < 0.01, OR 1.68, 1.2–2.35)), depth (*p* < 0.01, OR 1.61, 1.2–2.16), and stone location on CT (*p* = 0.01, ORs: kidney 19.1, 1.33–273, proximal ureter: OR 8.8, 1.04–73, middle ureter: 15.5, 1.75–136, distal ureter: OR 5.94, 0.74–47), reference category is intravesical ureter) were significant factors predicting the use of the laser. Multivariable analysis evaluating the duration of surgery at ≤30 min revealed that patient age ((*p* < 0.01, OR 1.017, 1.004–1.03), height (*p* = 0.03, OR 1.34, 1.23–1.45), depth (*p* < 0.01, OR 1.78, 1.54–2.05), width (*p* < 0.01, OR 1.70, 1.48–1.95), calculated spherical volume (*p* < 0.01, OR 1.0018, 1.0012–1.0026), and calculated cuboidal volume (*p* = 0.03, OR 1.003, 1.002–1.004) were significant factors evaluating surgery time. Stone composition has no effect on use of laser (*p* = 0.92, Pearson Chi-Square test) or time of surgery > 30 min (*p* = 0.92, Pearson Chi-Square test). Additionally, we added stone composition in both models to analyze the predictive power of stone composition if it is added to the other 10 predictive variables. Without using stone composition as predictor, model performance of the neural network for use of laser was 91% (Table 2). Then, we added stone composition as predictor, and retrained and reanalyzed the models. The result shows that model performance remained the same when stone composition is included (91%). Similar results were found for the time of surgery ≤30 min.

### 3.2. Performance of Other Machine Learning Algorithms

Support vector machines, nearest neighbor models, random forests, and the Bayes classifier showed similar results for both endpoints (Table 2) Model performances before applying the reject option were similar between support vector machines, random forests and neural networks. Proportions of unpredicted cases were also similar after applying the reject option. From our point of view, neural networks together with a reject option are most conveniently implemented in a computer system as compared to the other algorithms. Therefore, these models were further analyzed.

### 3.3. Model Performances of the Neural Networks

The best neuronal network model for the use of lithotripsy excluded 138 of 401 (34%) patients (Figure 2 and Table 3).

In the remaining 263 cases (66%), the corresponding negative predictive value was 97%, and the positive predictive value was 94% (Table 3 and Appendix A).

Positive predictive value was not computed, because the model excluded all patients with a duration of surgery >30 min. To provide a better understanding of the models, we provide real data (Table 4).

For explanation of why a certain number of patients received no prediction, we provide a deeper insight into how closely both data clouds (endpoint laser lithotripsy) are stuck together, illustrate the overlap between various data distributions and provide results based on PCA (Figure 1a,b). As illustrated, there are considerably large overlaps between various distributions, indicating the need for the reject option.

## 4. Discussion

Accurate prediction could be obtained using neural network modelling to identify patients undergoing ureterorenoscopy for the treatment of ureter stones not requiring laser lithotripsy and involving a short operation time (≤30 min). In more than half of all patients treated by ureterorenoscopy, this allows for a confident prediction of whether the procedure is a time-limited procedure without increased resource use. It has been reported that qualification of the surgeon influences operative time, requiring more complex calculation models. This has been shown for flexible ureterorenoscopy of renal calculi by Kuroda et al. In most cases, however, an exact number of minutes is not decisive, since the total duration of a procedure is also influenced by other factors such as anesthesia time. For daily planning, time slots are a practical and easy-to-use way to schedule surgeries. The more standardized a procedure is, the easier it is to predict. This is more applicable for semirigid ureterorenoscopy compared to flexible ureterorenoscopy. Thus, prediction of operative time in flexible ureterorenoscopy is more challenging because of a higher variability of technical aspects such as visibility issues, various stone location, use of ureteral access sheath, use of laser, complex handling of flexible instruments and surgeon’s expertise. It has been reported for laser lithotripsy that various factors such as stone size and density impacts delivered laser energy, thus influencing operative time. In addition to these variables, the prediction of laser use is an essential aspect. The information about the need of laser use affects not only operative time, but also the cost for consumables, surgical planning, resource allocation and laser availability.

Although advances in laser technology have made laser lithotripsy a safe and effective procedure for disintegrating stones, the intraoperative addition of the laser for lithotripsy results in longer operation time and more complex surgery compared to stone removal with baskets. Reliable prediction of the surgical time improves both the anaesthesia control time and patient scheduling, resulting in an improvement of the overall perioperative process. Thus, in the interdisciplinary surgical team, it is not only the surgeon who benefits from the preoperative knowledge of the laser application and the duration of the surgery. Using 34,976 cases over a 14-year period, Van Veen-Berkx et al. showed that more accurate scheduling of the two major components of a procedure (anaesthesia time and surgeon control time) resulted in fewer case cancellations, fewer prediction errors, and smoother workflow in the operating room [13]. Accurate prediction of surgical times is necessary for effective surgical planning. Solutions may include calculations from the expected operating time to account for the temporal impact of anaesthesia, rather than using a methodology to predict anaesthesia time based on a fixed number of minutes [14]. Thus, using a model to estimate the expected surgical time and using laser lithotripsy is helpful for potential calculations. It seems only logical that, through sensible planning of the procedure, reduction of the operation duration, and optimization of the internal hospital processes, a more efficient utilization of operating rooms occurs [15]. For example, a common cost driver is the elimination of elective surgeries that have to be canceled due to inefficient surgical scheduling. The reason for this, in addition to the occurrence of emergencies, is usually inefficient planning, which in turn arises due to a lack of information about the expected duration of the operation. This is where neural network prediction models can contribute. Furthermore, due to the short operation time and no need for laser lithotripsy, interventions that are suitable for colleagues in training can be identified. It should be mentioned that, besides stone location, size, and composition as some of the influencing factors, the stone treatment can proceed as either a simple stone removal or as a complex stone treatment. For example, stone composition is an important factor that can affect the duration of laser lithotripsy. Stones with high density usually require more intense energy delivery and longer lithotripsy. However, information about the exact composition is not available prior to surgery and therefore cannot be used to make a prediction. Our model shows high accuracy with the available preoperative information, such as the CT measured density in Hounsfield units, so that the lack of preoperative knowledge about the composition has no effect. Knowing whether a ureteral stone can be removed without further assistance from lasers is important to expedite intraoperative procedures and assess the complexity of the procedure given the skills of the surgeon. For the remaining patients for whom no prediction can be made, it is up to the surgeon to decide whether laser intervention is necessary and how long the procedure is likely to take. In this case, it depends on the assessment and experience of the surgeon or the surgical planner when evaluating the complexity of the procedure. The current analysis can impact the development of future prediction models integrating the entire endourological repertoire, thus improving surgical planning for complete endourological stone therapy. It would be useful to combine semirigid and flexible ureterorenoscopies to obtain a better overall picture of the whole endourological stone therapy. Irregularities in planning may still occur, which in turn may lead to the cancellation of operations or time overruns. Considering the high proportion of ureterorenoscopies for the treatment of urinary stones, time and cost savings are nevertheless achievable. Both models can be implemented in any database so that the predictions are done immediately and are ready at hand for the surgeon. This should lead to a more effective classification of ureterorenoscopic procedures to optimize resource allocation, reduce urologic case cancellations, and use hospital resources more efficiently.

## 5. Strengths and Limitations of This Study

### 5.1. Strengths

The most important strength of this study is that it provides prediction models for both primary endpoints with excellent accuracy to identify patients for which laser lithotripsy is not required and operation time is ≤30 min. This significantly improves the daily work of practical surgical planning. Additionally, the models are not only theoretical concepts but can be implemented in any data base and are of practical use for the clinic to improve planning of the surgical program and decrease costs. The large sample size of more than 400 patients is also an important strength of the study, because a large sample size is needed to arrive at statistically sound models, as emphasized by mathematical statistics and machine learning theory. It is important to train and independently validate the models in large samples to avoid over-learning and to provide accurate predictions when the models are confronted with new, previously unseen data (generalization to new data and model stability).

### 5.2. Limitations

The limitation of the study is the restriction to the analysis of only semirigid ureterorenoscopies. If a flexible ureterorenoscopy was performed intraoperatively due to proximal stone migration, these patients were not included in the study. However, due to the high number of cases in this study, high accuracy in prediction can be achieved once the model makes a statement for the initially planned semirigid ureterorenoscopies.

## 6. Conclusions

Taking into account the standard available patient data, more than half of ureterorenoscopic stone treatments are safely predicted for a duration of surgery ≤30 min and use of laser lithotripsy. Both prediction models significantly decrease the amount of practical surgical planning efforts in daily work, ensure appropriate utilization, and are ready to serve the surgery planner in scheduling surgeries more effectively.

## Figures and Tables

**Figure 1 jpm-12-00784-f001:**
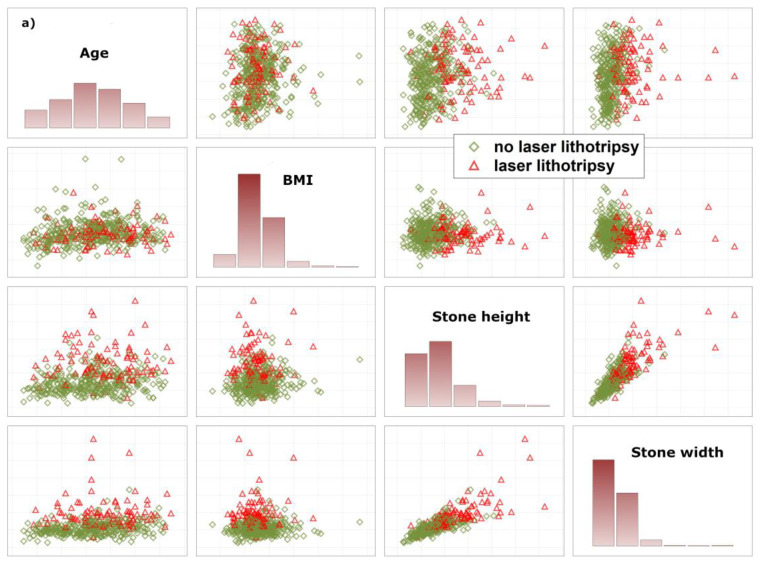
Illustration of the overlap between patients with and without laser lithotripsy. (**a**) illustrates raw data based on univariate variable combinations. (**b**) illustrates both data distributions based on principal component analysis. Both figures demonstrate the need for a reject option that is used to identify patients in the areas of overlap. These patients do not receive a prediction. Body Mass Index (BMI).

**Figure 2 jpm-12-00784-f002:**
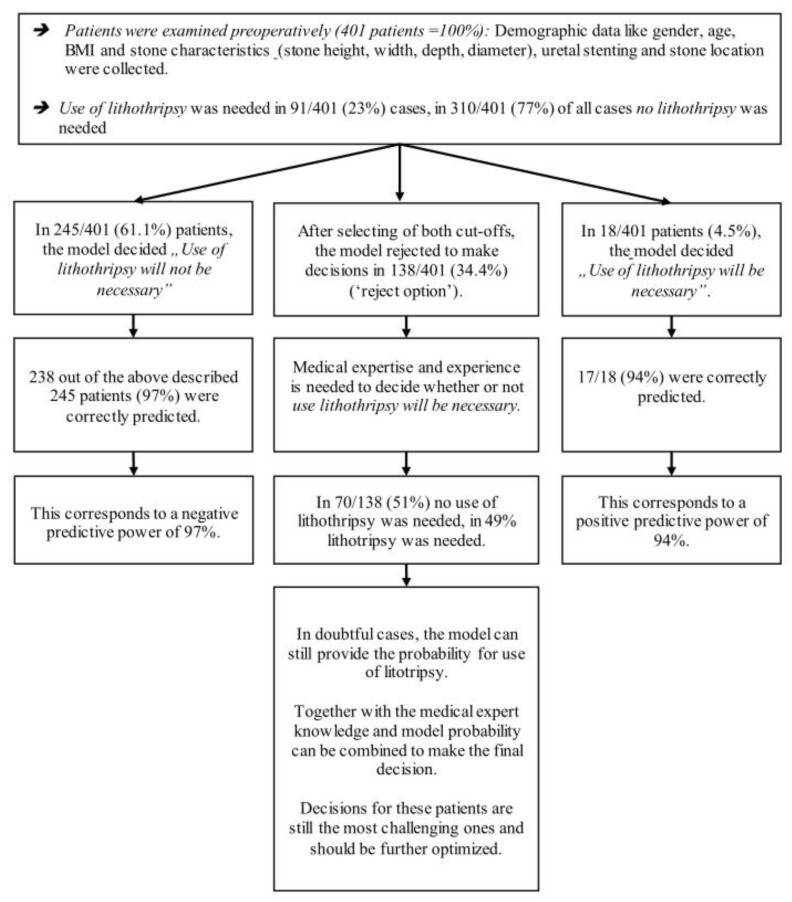
Illustration of model performance of the prediction model for use of lithotripsy.

**Table 1 jpm-12-00784-t001:** Overview of patient demographics.

Gender (male/female)	68%/32%
Age (mean/std)	53.6/16.7
Laser lithotripsy	22.5%
Stone removal by nitinol basket	77.5%
Ureteral stenting (yes/no)	74%/26%
Stone location (Kidney/proximal ureter/middle ureter/distal ureter/ ureter ostium)	12%/21%/19%/44%/4%
Time of surgery (min) (mean/std)	24.5/16.7
BMI (kg/m^2^) (mean/std)	27.6/5.3
Stone height (mm) (mean/std)	6.22/3.23
Stone width (mm) (mean/std)	4.89/2.48
Stone depth (mm) (mean/std)	5.22/2.39
Stone diameter (mm) (mean/std	6.58/3.36

**Table 2 jpm-12-00784-t002:** Overview of the results of five machine learning algorithms to predict use of lithotripsy and whether surgery ≤30 min. Model comparisons are based on negative and positive predictive values.

Results Are Based on 10-Fold Cross-Validation	Support Vector Machine	Nearest Neighbors	Random Forest	Bayes Classifier	Neural Network
Use of lithrotripsy (NPV/PPV)%	85/90%	88/86%	87/92%	63/93%	92/91%
Time surgery ≤ 30 min (NPV/PPV)%	87/83%	81/81%	78/86%	54/89%	63%/82%

Results before application of rejection option. For use of laser lithotripsy, model performances are very similar between support vector machines, random forests and neural networks. Negative Predicted Value, (NPV); Positive Predicted Value, (PPV).

**Table 3 jpm-12-00784-t003:** Overview of model performances after application of reject option to the neural network models for use of lithotripsy and time of surgery ≤ 30 min.

Use of Lithotripsy:				
	Negative Predictive Power	Positive Predictive Power	Unpredicted	Total Correctly Predicted
Training sample (10-fold cross-validation)	(214/220) 97%	(15/16) 94%	(124/360) 34%	(229/236) 97%
Test sample	(24/25) 96%	(2/2) 100%	(14/41) 34%	(26/27) 96%
Overall sample	(238/245) 97%	(17/18) 94%	(138/401) 34%	(255/263) 97%
Time of surgery ≤ 30 min:				
	Negative Predictive Power	Positive Predictive Power	Unpredicted	Total Correctly Predicted
Training sample (cross-validation)	(206/220) 94%	- ^1^	(140/360) 39%	(206/220) 94%
Test sample	(19/21) 91%	- ^1^	(20/49) 44%	(19/21) 91%
Overall sample	(225/241) 93%	- ^1^	(160/401) 40%	(225/241) 93%

^1^ For time of surgery, the positive predictive value was not computed, because the model made predictions only for those patients for which time of surgery is less than 30 min.

**Table 4 jpm-12-00784-t004:** Illustration of model performance to predict use of laser lithotripsy for seven patients. Green highlighted patients (columns) were correctly predicted, red highlighted patients were falsely predicted, and grey marked patients did not receive a prediction due the reject option. NPV and PPV were 92% and 91% before application of the reject option (Table 2) and increased to 97% and 94% after application of the reject option (Table 3). The cost for this improvement is that 34% of all patients (grey marked) did not receive a prediction.

	Pat ID	** * 1 * **	** * 2 * **	** * 3 * **	** * 4 * **	** * 5 * **	** * 6 * **	** * 7 * **
Age	years	** 42 **	** 80 **	** 52 **	** 37 **	** 90 **	** 88 **	** 62 **
BMI	kg/m^2^	** 30 **	** 28 **	** 23 **	** 27 **	** 23 **	** 28 **	** 31 **
Stone height	mm	** 15.5 **	** 6 **	** 9.9 **	** 6.9 **	** 13 **	** 4.7 **	** 5 **
Stone width	mm	** 8.7 **	** 5 **	** 6 **	** 4.9 **	** 11.9 **	** 3.3 **	** 3.4 **
Stone depth	mm	** 10.6 **	** 5 **	** 5.9 **	** 3.5 **	** 10.8 **	** 4.9 **	** 5 **
Stone diameter	mm	** 15.5 **	** 6 **	** 9.9 **	** 6.9 **	** 13 **	** 4.9 **	** 5 **
Hounsfield units		** 1600 **	** 570 **	** 1041 **	** 719 **	** 1259 **	** 680 **	** 567 **
Gender	m/f	** male **	** male **	** male **	** male **	** male **	** male **	** female **
Ureteral stenting	yes/no	** yes **	** yes **	** yes **	** yes **	** no **	** yes **	** yes **
Stone location ^1^	0–4	** 3 **	** 2 **	** 1 **	** 1 **	** 0 **	** 4 **	** 4 **
Observed use of laser	laser: yes/no	** yes **	** no **	** no **	** no **	** yes **	** no **	** no **
Predicted use of laser	laser: yes/no	** no prediction **	** no **	** no prediction **	** no **	** no **	** no **	** no **

^1^ Stone location: 0 = Kidney, 1 = Proximal ureter, 2 = Middle ureter, 3 = Distal ureter, 4 = Intravesical ureter.

## Data Availability

The datasets used and/or analysed during the current study are available in anonymised form from the corresponding author on reasonable request.

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
