# Peer review of "Neural Networks Modeling for Prediction of Required Resources for Personalized Endourologic Treatment of Urolithiasis"

_jpm, 2022, doi:10.3390/jpm12050784_

Round 1
Reviewer 1 Report
This is a retrospective study of 474 patients who underwent URS between 2016 and 2019. The authors used machine learning models and reported > 90% of NPV and PPV to predict the need for laser lithotripsy and expected duration of surgery at < 30 min.
Herein are my questions and comments:
- The introduction should include a summary of previous studies on this topic to justify the research question. There are several papers on this topic. For example: Kuroda S, et al. A new prediction model for operative time of flexible ureteroscopy with lithotripsy for the treatment of renal stones. PLoS One. 2018;13:e0192597; Ofude M, et al Stone Attenuation Values Measured by Average Hounsfield Units and Stone Volume as Predictors of Total Laser Energy Required During Ureteroscopic Lithotripsy Using Holmium:Yttrium-Aluminum-Garnet Lasers. Urology. 2017;102:48-53.
- The authors included “all patients who had semirigid URS with a single stone in the ureter or pelvis of the kidney”. However, semirigid URS is not feasible and indicated in “kidney” stones. Also, they then stated that flexible URS was used to treat renal pelvis and calyceal system. This brings some confusion regarding the inclusion criteria.
- How did the authors come up with a cutoff of “30 min” for the operative time? Are there any other studies or prelim data that justify this limit?
- The authors should elaborate on the pre-op evaluation and intra-op technical points. Did CT scan obtain for all patients before surgery? What was the indication for JJ placement?
- For multivariable analysis (lines 166-176), the odds ratios of the predictors should be reported.
- While reporting BMI as a surrogate for body habitus, including height and weight in the analytical model is not necessary. The same issue for reporting both stone “volume” and stone “diameters”.
- Is there any data available for the stone-free and pushback rates?
- The discussion has the same issue as comment 1. Relevant papers should be discussed in this section.
Author Response
Dear Reviewer,
Thank you for your detailed and helpful review. We have answered to all raised questions point by point and have processed all your suggestions. The manuscript could be improved substantially and therefore, we hope the manuscript will be suitable for publication.
Reviewer Comments:
- The introduction should include a summary of previous studies on this topic to justify the research question. There are several papers on this topic. For example: Kuroda S, et al. A new prediction model for operative time of flexible ureteroscopy with lithotripsy for the treatment of renal stones. PLoS One. 2018;13:e0192597; Ofude M, et al Stone Attenuation Values Measured by Average Hounsfield Units and Stone Volume as Predictors of Total Laser Energy Required During Ureteroscopic Lithotripsy Using Holmium:Yttrium-Aluminum-Garnet Lasers. Urology. 2017;102:48-53
Author reply: Thank you very much for this helpful and useful note to include additional literature to further explain the importance of this topic. We have now included additional sources in the manuscript to highlight the importance of surgical scheduling and operative times for ureterorenoscopy procedures.
Author action:
We have included the following section in the introduction to provide a better understanding of the context:
There are already isolated attempts to simulate the interventions and their duration for better planning. For the treatment of kidney stones using flexible ureterorenoscopy with laser lithotripsy, prediction models have been developed to predict the operation time. It was shown that different parameters, such as the presence of a ureteral stent before surgery or gender, can influence the operation time for flexible uretereorenoscopy. Using different preoperative data, models for calculating the operation time can thus already be created for flexible uretereorenoscopy. The more complex the procedure, the more factors naturally play a role in accurately predicting the operating time. It is known that preoperative parameters such as stone density and stone size have an influence on the laser energy used in the case of laser application. Thus, there is also an effect on the operation time due to the duration of the energy applied. However, prediction models regarding the operation time should at best determine the requirement for laser application in advance in order to allow for better planning of the operation. (Page 2 of 12)
We added the word semirigid in the last sentence and have added the relevant references:
- The authors included “all patients who had semirigid URS with a single stone in the ureter or pelvis of the kidney”. However, semirigid URS is not feasible and indicated in “kidney” stones. Also, they then stated that flexible URS was used to treat renal pelvis and calyceal system. This brings some confusion regarding the inclusion criteria.
Author reply: Thank you for pointing this out. We understand that this point can lead to confusion and also see the need to change the wording. In addition to ureteral stones, stones located in the pyeloureteral junction can sometimes be accessed and removed by semirigid ureterorenoscopy without the need for extra sheath insertion and flexible ureterorenoscopy. Later we describe, that patients with nephrolithiasis of the renal pelvis or calyces that could only be treated by flexible ureterorenoscopy were not included
Author action: We have edited the following sentences:
All patients who underwent semirigid ureterorenoscopy with a single stone in the ureter or through semirigid accessible portion of the renal pelvis (pyeloureteral junction) were included.
All stones that could be treated by semirigid ureterorenoscopy were included which reflects the clinical reality most appropriately. (Page 2 of 12)
- How did the authors come up with a cutoff of “30 min” for the operative time? Are there any other studies or prelim data that justify this limit?
Author reply: This is an important issue, which we discussed extensively. As described, we opted for the 30 minute slots, since the semi-rigid ureterorenoscopy is a manageable, standardized procedure. Data from Mahmood et al. show that flexible ureterorenoscopy with an operation time > 30 min can lead to increased complications. In a study by Reicherz et al. patients with a mean operation time of 21.1 min were treated without complications using short-term Mono-J insertion. Even though the authors describe that the operating time generally had no influence on the complication rate for short-term Mono-J insertion, the limit of 30 min is realistic as a reproducible time window for semirigid uretereorenoscopy. Taking into account the time required for anesthesia, intervention times in clinical practice not exceed a total of 1 hour. This makes the operation planner's work easier and enables clear time slots to be created for the semirigid ureterorenoscopies – we have also explained it this way in the manuscript. Of course, delays in anesthesia or unforeseen complications during anesthesia cannot be represented by the prediction model.
Mahmood SN, Toffeq H, Fakhralddin S. Sheathless and fluoroscopy-free retrograde intrarenal surgery: An attractive way of renal stone management in high-volume stone centers. Asian J Urol. 2020 Jul;7(3):309-317. doi: 10.1016/j.ajur.2019.07.003. Epub 2019 Jul 16. PMID: 32742931; PMCID: PMC7385507.
Reicherz A, Maas V, Reike M, Brehmer M, Noldus J, Bach P. Striking a balance: outcomes of short-term Mono-J placement following ureterorenoscopy. Urolithiasis. 2021 Dec;49(6):567-573. doi: 10.1007/s00240-021-01264-4. Epub 2021 Apr 13. PMID: 33847780; PMCID: PMC8560726.
Author action: We added following sentence and updated the references:
Studies have also shown that patients with ureterorenoscopy that lasted less than 30 minutes, postoperative short-term Mono-J insertion is a possible option without the risk of additional complications. (Page 3 of 12)
- The authors should elaborate on the pre-op evaluation and intra-op technical points. Did CT scan obtain for all patients before surgery? What was the indication for JJ placement?
Author reply: Thank you for this valuable suggestion. We included the information regarding the CT scans and the indications for JJ placement in the manuscript.
Author action: We have added the following sentences to the manuscript for better understanding:
Inclusion criteria was the presence of CT imaging, either as part of the emergency initial presentation or at time of preoperative presentation. (Page 2 of 12)
In case of preoperative ureteral stenting, this was placed during the emergency first visit due to therapy-refractory colic or septic constellation. (Page 3 of 12)
- For multivariable analysis (lines 166-176), the odds ratios of the predictors should be reported.
Author reply and action: Odds ratios with 95% CI for the predictors are inserted now.
- While reporting BMI as a surrogate for body habitus, including height and weight in the analytical model is not necessary. The same issue for reporting both stone “volume” and stone “diameters”.
Author reply: Both stone “volumes” (spherical and cuboid), body height and weight are removed from the models now. BMI is still included. We retrained and reevaluated all models after inclusion and exclusion of stone diameter and we found stone diameter still useful. After exclusion of stone volumes, body height and weight, model performances remained almost identical (see Table 3, Figure 2).
Author action: Both stone “volumes”, body height and weight were removed from the models now. BMI is still included. Results were updated in the results section and were also updated in Table 3, Table 4, Figure 2 and Figure S1 (supplementary material).
- Is there any data available for the stone-free and pushback rates?
Author reply: All information on the stone-free rates at the end of the operation is available in our data. In 5 patients there was no stone-free after the procedure, of which 2 stones were larger than 15 mm.
The pushback rates during the operation were not recorded for the creation of the prediction model. Pushback rates were not documented. Please note that the prediction for the individual patients has to be made by the model before semirigid ureterorenoscopy will be performed. So pushback rates were not included in the model for prediction. Please also note that the model was trained and tested with patients with and without possible pushbacks that occurred during surgery.
- The discussion has the same issue as comment 1. Relevant papers should be discussed in this section.
Author reply: Thank you very much for your comment. We have now included the relevant papers in the discussion section.
Author action: Following sentences are now included in the manuscript:
It has been reported that qualification of the surgeon influences operative time requiring more complex calculation models. This has been shown for flexible ureterorenoscopy of renal calculi by Kuroda et al. In most cases, however, an exact number of minutes is not decisive, since the total duration of a procedure is also influenced by other factors such as anesthesia time. For daily planning time slots are a practical and easy-to-use way to schedule surgeries. The more standardized a procedure is, the easier it is to predict . This is more applicable for semirigid uretereorenoscopy compared to flexible uretereorenoscopy. Thus, prediction of operative time in flexible ureterorenoscopy is more challenging because of a higher variability of technical aspects such as visibility issues, various stone location, use of ureteral access sheath, use of laser, complex handling of flexible instruments and surgeon´s expertise.
It has been reported for laser lithotripsy that various factors such as stone size and density impacts delivered laser energy, thus influencing operative time. In addition to these variables, the prediction of laser use is an essential aspect. The information about the need of laser use impacts not only operative time, but also influences costs for consumables, surgical planning, resource allocation and laser availability. (Page 9 of 12)
I hope we were able to answer all your questions and improve the article with your expertise. I am looking forward to answer any questions and hope that the revision will find your acceptance.
Reviewer 2 Report
I believe that diagnosing the need for laser lithotripsy with AI, as discussed in this paper, would be beneficial for surgical planning.
We want to receive comments on the following.
1) How much clinical impact do you see in preoperative planning? How much loss amount?
2) Regarding Table 2, you state that the need for lithotripsy is 84-90%, and the time is 78-85% predictive accuracy. How much improvement would be possible with these results?
3) What does it mean to say that ureteroscopy was done but not lithotripsy in Materials and Methods? Wouldn't it be better to add data from patients with spontaneous stone removal to the analysis?
4) In this study, only a rigid ureteroscopy is used, but it may be safer to push up into the renal pelvis for lithotripsy using flexible ureteroscopy in clinical practice. Isn't it also necessary to use data that also used a flexible ureteroscopy?
5) The subject physicians are 17 urologists; how did you evaluate the skill of the surgeons? Are there differences in the experience?
I think that surgery by an experienced urologist will take less time. How much did differences in experience affect your results?
6) Is it the size of the stone that has the most significant impact on the outcome? If so, it does not seem very novel.
7) Did the type of stone component make a difference?
8) How long is the ureteral stent in place? I would think that the longer the indwelling period, the easier it would be to drain the stone.
9) Do you consider surgery time by the site of the stone?
Lower ureteral stones are easier to do, but UVJ stone is harder.
10) Are cases like duplicated ureters and horseshoe kidneys included?
11) Is there any comparison of the parameters of the results?
Author Response
Dear Reviewer,
Thank you for your detailed and helpful review. We have answered to all raised questions point by point and have processed all your suggestions. The manuscript could be improved substantially and therefore, we hope the manuscript will be suitable for publication.
Reviewer Comments:
- How much clinical impact do you see in preoperative planning? How much loss amount?
Author reply: This is an interesting point and we thank the reviewer for addressing this issue. Procedures can be precisely predicted with respect to short operative time (<30 min) and need for laser use in more than half of the patients. For these patients, effective surgical planning can be performed. On the other hand, complex procedures can be identified and filtered out avoiding surgical cancellations and postponements In addition, surgical planning is complex and changing permanently due to ermergency cases and unforeseen circumstances in surgery. The use of prediction models, as reported by Kuroda et al. for flexible ureterorenoscopy, the number of endourological stone treatments can be planned more effectively. For predicted complex procedure, experienced surgeons can be planned to reduce operative times.
- Regarding Table 2, you state that the need for lithotripsy is 84-90%, and the time is 78-85% predictive accuracy. How much improvement would be possible with these results?
Author reply: Please note that the intention of Table 2 was to demonstrate a more technical aspect of the paper showing that various statistical models are leading to similar results. Tab. 2 suggests that the model building process may not be further improved by using other models, since the most common statistical models perform very similar. To answer your question how much improvement would be possible, we refer to the discussion where we have outlined important improvements in daily work of practical surgical planning.
Author action: Please find the following paragraphs in the discussion section:
In more than half of all patients treated by ureterorenoscopy, this allows for a confident prediction of whether the procedure is a time-limited procedure without increased resource use.
Although advances in laser technology have made laser lithotripsy a safe and effective procedure for disintegrating stones, the intraoperative addition of the laser for lithotripsy results in longer operation time and more complex surgery compared to stone removal with baskets. Reliable prediction of the surgical time improves both the anaesthesia control time and patient scheduling, resulting in an improvement of the overall perioperative process. Thus, in the interdisciplinary surgical team, it is not only the surgeon who benefits from the preoperative knowledge of the laser application and the duration of the surgery. (Page 9 of 12)
- What does it mean to say that ureteroscopy was done but not lithotripsy in Materials and Methods? Wouldn't it be better to add data from patients with spontaneous stone removal to the analysis?
Author reply: This prognostic model focuses on predicting operative time and laser use in semirigid ureterorenoscopies for stone treatment, patients who did not require laser for stone removal can be identified. This of course includes patients with small calculi that could be easily removed by nitinol basket or patients with spontaneous stone passage in the meantime. Please note that patients with spontaneous stone removal are already included in the training and test samples. The decisive aspect is that the less complex and shorter 30 min. operations can identified in this way, which has an effect on the planning of operations.
- In this study, only a rigid ureteroscopy is used, but it may be safer to push up into the renal pelvis for lithotripsy using flexible ureteroscopy in clinical practice. Isn't it also necessary to use data that also used a flexible ureteroscopy?
Author reply: We would like to thank the reviewer for clarifying this issue.
In our clinical practice, in about 25% of cases proximal ureter stones are pushed back into the renal pelvis requiring flexible ureteroscopy. However, we have excluded this group of patients from calculation because we wanted to focus on a standardized procedure for our prediction model. However, the next step is now to integrate all endourologic stone treatment (rigid and flexible ureteroscopy) into these models.
Author action: We added the sentence to discussion: The current analysis can impact the development of future prediction models integrating the entire endourologic repertoire, thus improving surgical planning for complete endourological stone therapy.it would be good to combine semirigid and flexible ureterorenoscopies to get a better overall picture of the whole endourological stone therapy. (Page 10 of 12)
- The subject physicians are 17 urologists; how did you evaluate the skill of the surgeons? Are there differences in the experience? I think that surgery by an experienced urologist will take less time. How much did differences in experience affect your results?
Author reply: Thank you for the important question. For reasons of quality assurance and standardization, only interventions by specialists were included in the creation of the prediction model. This means that a minimum of 50 endourological treatments on the ureter are required. Of course, there were differences in experience within the group of specialists. The fact that we have worked with a total of seventeen specialists to create the model allows us to represent a broad range of urological expertise in semirigid ureterorenoscopy. In our opinion, this is the most realistic representation of clinical practice and ensures stability of the model perfomances when other urologists performing semirigid ureterorenoscopies.
Author action: We have made changes in the Material and Methods and included the sentence:
For reasons of quality assurance only interventions by specialized urologists were included in the creation of the prediction model. Procedures performed by urologists who had less expertise (< 50 cases of rigid ureteroscopy) were excluded from the analysis. Therefore, procedures from 17 surgeons were included into the study allowing a broad representation of the clinical urological expertise when creating the prediction model and ensuring stability of the model. (Page 3 of 12)
- Is it the size of the stone that has the most significant impact on the outcome? If so, it does not seem very novel.
Author reply: We would like to thank the reviewer to clarify this issue.
The focus of the manuscript was to predict operative time and the need of laser use for lithotripsy. The size of the stone as well as the location of the stone have significant impact the outcome.
To clarify the impact of predictors, we have included the odds ration of the predictors into the manuscript. However, it is still very difficult to assess which predictor has the most significant impact on the outcome. It is well known and well described in mathematical/statistical literature that a ranking of predictor importance has many pros and cons and that a ranking itself is not always possible. Please note that it was not the aim of this study to assess which predictor has the most significant impact on the outcomes. Thus, we are cautious here and make no statements on this important and difficult issue. We think a follow-up study should make comprehensive and extensive statistical analyses based on modern state of the art feature selection and ranking methods to find a better understanding of predictor importance.
- Did the type of stone component make a difference?
Author reply and action: Stone composition has no effect on use of laser (p=0.92, Pearson Chi-Square test) or time of surgery >= 30 minutes (p=0.92, Pearson Chi-Square test). Additionally, we added stone composition in both models to better understand the predictive power of stone composition if it is added to the other 10 predictive variables. Without using stone composition as predictor, model performance of the neural network for use of laser was 91% (Tab.2). Then, we added stone composition as predictor, retrained and reanalyzed the models. The result shows that model performance remained the same when stone composition is included (91%). Similar results were found for the time of surgery <= 30 min.
We added the sentences at results section. (Page 6 of 12)
- How long is the ureteral stent in place? I would think that the longer the indwelling period, the easier it would be to drain the stone.
Author reply: If ureteral stenting was present before ureterorenoscopy, the interval to surgery was on average 2 weeks. You are absolutely correct that pre-stenting increases stone-free rates. This was also shown in a systematic review by Yang et al. There were no significant differences for the duration of surgery. Our prediction model makes the decision whether laser intervention is necessary in the same way as for surgery time. It uses the information of an existing pre-stenting. In this case, the time does not have to be included in the model, since no changes in the operation times are to be expected.
Yang Y, Tang Y, Bai Y, Wang X, Feng D, Han P. Preoperative double-J stent placement can improve the stone-free rate for patients undergoing ureteroscopic lithotripsy: a systematic review and meta-analysis. Urolithiasis. 2018 Oct;46(5):493-499. doi: 10.1007/s00240-017-1012-z. Epub 2017 Nov 1. PMID: 29094191.
- Do you consider surgery time by the site of the stone?
Lower ureteral stones are easier to do, but UVJ stone is harder.
Author reply: Thank you for addressing important issue with semirigid ureterorenoscopy. We agree that distally located stones are easier and faster to extract. We took into account that stone location is an important predictor. Therefore, we included the preoperatively known stone location from CT scan as a predictor for our model. Please note that time of surgery can not be used as a predictor because the prediction has to be done before finale ureterorenoscopy will take place.
- Are cases like duplicated ureters and horseshoe kidneys included?
Author reply: Anomalies such as horseshoe kidney or ureter duplex were excluded.
Author action: We have added the information to the inclusion and exclusion criteria. (Page 2 of 12)
- Is there any comparison of the parameters of the results?
Author reply: Please note that the whole sample was split into a training and test sample. A comparison of model performances was done of the results in the training sample and test sample. Please note total correctly predicted patients were 97%/94% for use of lithotripsy/time of surgery <= 30’ in the training sample. In the test samples, corresponding numbers were 97%/93% as shown in Tab.3. This comparison demonstrates that the models perform well when confronted with new, but previously unseen patients.
I hope we were able to answer all your questions and improve the article with your expertise. I am looking forward to answer any questions and hope that the revision will find your acceptance.
Round 2
Reviewer 1 Report
The authors have appropriately addressed all the comments and questions. I have no further comments.
Author Response
Manuscript: jpm-1664710
Dear Reviewer,
Thank you for your positive review. We performed an additional English spell check and by revising the manuscript with your support, we hope that the manuscript will be suitable for publication.
Reviewer Comments:
The authors have appropriately addressed all the comments and questions. I have no further comments
Author reply: Thank you very much your positive feedback.
We still looking forward to answer any questions and hope that the revision will find your acceptance.

Reviewer 2 Report
I think the author has answered the reviewers' questions accurately.
How do you assess the validity of the statistical analysis methods? I am not an expert in statistics, so could you please enlighten me?
Author Response
Manuscript: jpm-1664710
Dear Reviewer,
Thank you again for your review and your question. We have answered your final question and have performed an additional spell check. By answering your question and revising the manuscript with your support, we hope that the manuscript will be suitable for publication.
Reviewer Comments:
How do you assess the validity of the statistical analysis methods? I am not an expert in statistics, so could you please enlighten me?
Author reply: Thank you for this particularly important question. Assessment of the validity of the statistical methods in context of machine learning refers to the following question:
“Will a machine learning model work when confronted with new, previously unseen patients?”
In statistical learning theory, it is well known that it is easy to build a model that performs well on the data on which the model has already learned – but fails when confronted with new, previously unseen patients.
In other words: Model validation refers to the process where the trained model is evaluated with new test data. The main purpose of model validation is to test the generalization ability of a trained model.
To answer your question: to check the generalization ability of the models, we have applied 2 steps:
- Data were split into learning, validation and test samples. In the learning sample, the model extracts the information from the data and learns as much as possible. In the validation samples, various internal parameter of the model were adapted. This was based on so-called 10-fold cross-validation. After this step, decision cut-offs were selected and the final model was fixed.
- After step 1, the test sample was used to independently test the model based on new and previously, unseen patients. It is important to note that the trained model has never “seen” the data of the test sample before. To assess model validity, model performances were analyzed in all samples and the results show that the model performed stable in all samples which shows that the model generalizes to new, previously unseen patients.
We hope you will find our answer suitable and that our reply contributes to a better understanding of the importance of model validity.
We still looking forward to answer any questions and hope that the revision will find your acceptance.

This manuscript is a resubmission of an earlier submission. The following is a list of the peer review reports and author responses from that submission.
Round 1
Reviewer 1 Report
Dear Author,
I have reviewed the article entitled "Neural networks modeling for prediction of required resources for personalized endourologic treatment of urolithiasis". Although the main idea of ​​the study is well thought out, there are methodological errors. Study must be re-designed. I have some comments below;
- I could not understand why the felxible ureerorensocopy is excluded? Current guidelines of European Association f Urology (EAU) about urolithiasis is recommend flexible ureterorenscopy for upper urinary tract system and kidney stoned smaller than 2 cm, not recommend to use rigid or semirigid ureterorenoscopy. So that, you must use flexible ureoterorenoscopy for the treatment of these stones. Please explain this.
- In material and method section, The laser size (272 nm, 550 nm, etc) must be clarify, and also energy source (holmiu laser ,Thillium laser, pneumotic, electrohydrolic,...???) to fragmented the stone must be clarify. Additionally what energy (30W, 45W or 60W,etc ???) is used?These parameters are so important and related with the operation time. Please clarify the lser size and energy source.
- I think, another limitation of the study is absence of the data of stone composition. This is another related risk factor that may efext operation time. Fragmantation time of different kind of stones are different. Please clarify the stone composition or add this parameter as a limitation factor in discussio section
- Preoperaive previous stone treatment is not explaned. I suggest that patients had preoperative extracorporeal shock wave lithotripsy must be excluded. Because this treatment may affect operastion time due to eudema or other factor. please clarify this
- Cut-off size of stones for the treatmen of ureterorenoscopy must be explain in material an method section. This is the most important risk facotr for stone fragmentation and operation time. Which patients included the study according to the stone size?
- table-1 must be re-desinged. In the laser lithotripsy yes/no line, laser treaten patients are too low. Please explaine how the patients with stones that not treated with laser lithoripsy in material and method section?
- Results of the study both given in table-1 and in the pragraph. Please give one of them, not duplicate your results
- how many surgeon perormed the ureterorenoscopy procedure? This is another problem that was related wtih the experience of surgeon and operastion time? please clarify
- Kidney stones must be explain more detail. Becuse calyxeal stones con not be treated eith rigid or semirigid ureterorenoscopy, it needs flexible ureterorenoscopy. please explain this
- in discussion section; line 202-205; I think sentences start wtih "intraoperastive addition of the laser for lithotripsy....." could be removed. This is not true. Due to developing technology, laser is the most effective treatmen of stone disease. It is easy, safe and effective. Or you must add references for this.
Best regards
Reviewer 2 Report
The criteria for including some renal pelvis stones in study , in order to treat them with rigid or semirigid ureterorenoscope should be defined .
Reviewer 3 Report
I’m thankful to the Authors for their interesting article entitled ‘Neural networks modeling for prediction of required resources for personalized endourologic treatment of urolithiasis’ for the Journal of Personalized Medicine. They aimed to develop and train the prediction models for ureterorenoscopy for optimising surgical planning and resource allocation to increase efficiency
The article is well-written, the purpose is clear. Relevant references were used for the whole article. The design and methods are clear. Results were presented clearly, with tables and figures. Strengths and limitations are clear. The conclusion was supported with the purpose and results of the study.